# Gait Recognition in Large-scale Free Environment via Single LiDAR

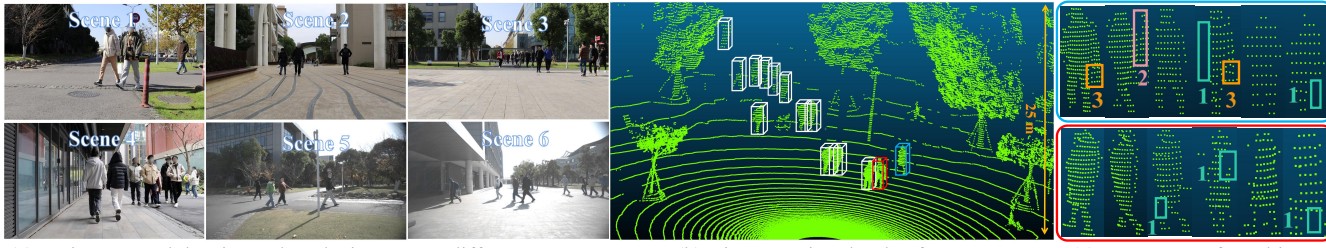

(a) various crowd density and occlusion across different scenes     (b) LiDAR point clouds of scene 3     (c) sequences of 2 subjects

**Figure 1: We propose FreeGait, a new LiDAR-based in-the-wild gait dataset under various crowd density and occlusion across different real-life scenes. FreeGait is captured in diverse large-scale real-life scenarios with free trajectory, resulting in various challenges such as 1. occlusions, 2. noise from crowd, 3. noise from carrying objects and etc., as shown in the right part.**

## ABSTRACT

Human gait recognition is crucial in multimedia, enabling identification through walking patterns without direct interaction, enhancing the integration across various media forms in real-world applications like smart homes, healthcare and non-intrusive security. LiDAR's ability to capture depth makes it pivotal for robotic perception and holds promise for real-world gait recognition. In this paper, based on a single LiDAR, we present the Hierarchical Multi-representation Feature Interaction Network (HMRNet) for robust gait recognition. Prevailing LiDAR-based gait datasets primarily derive from controlled settings with predefined trajectory, remaining a gap with real-world scenarios. To facilitate LiDAR-based gait recognition research, we introduce *FreeGait*, a comprehensive gait dataset from large-scale, unconstrained settings, enriched with multi-modal and varied 2D/3D data. Notably, our approach achieves state-of-the-art performance on prior dataset (SUSTech1K) and on *FreeGait*. **Code and dataset will be released upon publication of this paper.**

## CCS CONCEPTS

• **Computing methodologies** → **Biometrics**; **Activity recognition and understanding**.

## KEYWORDS

Gait Recognition, Human-centric Multimedia Understanding, LiDAR point cloud

*ACM MM, 2024, Melbourne, Australia*
© 2024 Copyright held by the owner/author(s). Publication rights licensed to ACM.
ACM ISBN 978-x-xxxx-xxxx-x/YY/MM
https://doi.org/10.1145/nnnnnnn.nnnnnnn

**Unpublished working draft. Not for distribution.**

## 1 INTRODUCTION

Gait is a promising biometric feature for human identification [4, 8, 9, 12, 14, 19, 22] that is difficult to disguise and can be captured from a distance without intrusive interactions. This characteristic is particularly significant for multimedia applications [21, 39, 42, 46, 50], where the seamless integration of information across various media forms is paramount. For example, in smart homes and IoT environments, gait recognition tailors user experiences by automatically adjusting environmental controls to individual preferences based on their movement patterns. Gait recognition also provides advanced diagnostic and monitoring tools in healthcare, offering insights into patient health and flagging potential issues, thus integrating essential health data with multimedia systems for improved patient management. Additionally, its non-intrusive nature makes gait recognition ideal for security and surveillance, enabling passive monitoring in environments where conventional methods may be too invasive. Despite being a subject of extensive research for decades, human gait recognition continues to be highly relevant and holds significant potential in the evolving multimedia landscape.

Existing large-scale LiDAR-based gait datasets [31] are predominantly collected within controlled lab environments where subjects follow a predetermined routine in a limited space. This lab setting often prompts participants to walk in a manner they believe is expected, potentially skewing their natural gait. Instructions to walk along a straight line, maintain a certain speed, or repeatedly traverse a marked path can render the collected data artificial, contrasting sharply with the spontaneous and varied movements typical in real-world scenarios. This discrepancy creates a gap between lab-collected gait data and the naturalistic human gait in complex real environments.

To bridge this gap in human gait recognition, we introduce **FreeGait**, a comprehensive dataset captured in open public areas such as subway exits, school gates, and sidewalks (see Figure 1). This dataset encompasses 1,195 subjects of diverse ages and genders, all walking freely in large-scale, unconstrained settings that reflect true pedestrian behavior in both sparse and crowded conditions.

Besides 2D silhouettes and 3D point clouds, FreeGait provides diverse data representations, including 2D/3D poses and 3D Mesh & SMPL models, fostering extensive research opportunities.

FreeGait enhances the gait dataset landscape with several novel features:

- **Naturalism**: Data captured from real pedestrians in open public spaces showcases more authentic walking behaviors than lab-based datasets.
- **Diversity**: The dataset features a broader demographic spectrum, including varied ages, genders, clothing preferences, etc.
- **Environmental Variability**: It captures variations in lighting, weather, and other conditions, crucial for developing robust, adaptable gait recognition algorithms.
- **Complex Interactions and Behaviors**: The real-world setting of our dataset allows for the observation of complex behaviors such as navigating obstacles, walking in groups, or carrying load. These real-world noises often absent in lab data but vital for nuanced gait analysis.

Ethical and privacy considerations were paramount in the development of FreeGait. We anonymized all subjects by blurring faces and secured post-event informed consent to ensure ethical integrity in our research.

Prevailing camera-based methods [4, 8, 9, 12, 19, 22, 35] encounters performance bottlenecks due to limited 2D visual ambiguities with view-dependent, illumination-dependent, and depth-missing properties of images. While RGB-D camera-based methods[11] have been proposed to extract more depth information for gait features, their limited range and inability to function outdoors hinder their applications. In contrast, LiDAR can capture accurate depth information in large-scale indoor and outdoor scenes, unaffected by light conditions. LiDAR point clouds of subjects represent genuine gait-related geometric and dynamic motion attributes, definitely benefiting gait-based identification. A pioneering LiDAR-based gait recognition study, LiDARGait[31], employs range view projected from point clouds, demonstrating the capability and effectiveness of LiDAR. However, its dimensionality reduction inevitably diminishes the undistorted geometric and dynamic gait information in the original 3D point clouds.

Recognizing that dense and regular range views projected from LiDAR point clouds enhance human body structure extraction, which are what sparse and unordered raw point clouds lack, we present a gait recognition approach using projected range view and raw point cloud representations. It can be applied in large-scale scenarios and varying light conditions, making it practical for intelligent security and assistive robots. We introduce the Hierarchical Multi-representation Feature Interaction Network, dubbed **HMRNet**, that synergizes the dense range view with raw point clouds. The range branch focuses on dense and regular body structure, while the point branch extracts the geometric information and explicitly models gait-related motion through our motion-aware feature embedding. Given huge domain gap between two LiDAR representations, we design an adaptive cross-representation mapping module to effectively fuse their features. By extracting multi-resolution features, we ensure both local fine-grained and global

semantic gait features are captured. After obtaining comprehensive fusion gait features, our gait-saliency feature enhancement module, utilizing a channel-wise attention mechanism, enhance gait-informative features for precise identification.

The main contributions can be summarized below.

- Based on a single LiDAR, we propose a novel gait recognition method that effectively captures both dense body structure features from LiDAR range views and gait-related geometric and motion information from raw point clouds, which is applicable for real-world scenarios without light and view constraints.
- We propose a large-scale in-the-wild free-trajectory gait recognition dataset with diverse real-world scenarios, data modalities, and data representations, which can facilitate the gait community to conduct rich exploration.
- Our method achieves state-of-the-arts performance on previous LiDAR-based gait dataset (SUSTech1K) and our FreeGait.

## 2 RELATED WORK

**Gait Recognition Methods.** Previous gait recognition methods [17, 21, 36, 39, 42, 46, 50] primarily use 2D image representation and fall into appearance-based and model-based approaches. The former [4, 8, 9, 12, 14, 19, 22] relies on silhouettes from RGB images, making performance dependent on segmentation quality and causing difficulty in identifying humans with changed clothes or cross views. Model-based methods [20, 29, 35] employ skeletons to capture genuine gait characteristics, but are heavily limited by model-based estimation methods. To address depth ambiguity in 2D, some methods [3, 11, 27, 44] employ RGB-D cameras for 3D gait features. Yet, depth cameras are limited to indoor scenes with confined sensing range. Recent 3D representation-based approaches[49] generate 3D meshes from RGB images, but are sensitive to quality and can introduce cumulative errors. [1, 2] explore view-robust gait recognition frameworks based on LiDAR point cloud. However, they all project 3D point cloud into 2D depth map for feature extraction, losing original 3D geometric information and resulting in limited performance. While LiDARGait [31] shows promising results by projecting LiDAR point clouds into dense range views, it still lacks essential geometric and dynamic information from raw point clouds. Our approach synergizes range views with 3D point clouds for superior gait recognition.

**Gait Recognition Datasets.** Existing open gait datasets [15, 18, 26, 32, 34, 40, 45, 51] primarily use silhouettes and are limited to controlled environments, making them unsuitable for real-world applications. GREW[51] addresses this by collecting a dataset in an open area for practical applications. Gait3D[49] attempts to solve the distortion problem in view-dependent images, but the auxiliary 3D mesh from RGB images still cannot represent the real depth and 3D motion properties. SZTAKI-LGA[2] collects a LiDAR-based gait dataset, capturing accurate depth information in large-scale scenes, but includes only 28 subjects and limited sequences, insufficient for evaluating learning-based approaches. A recent work introduces a LiDAR-based gait dataset, SUSTech1K[31], with 1,050 subjects. However, it is captured in constrained environments with predefined routes, creating a gap compared to real-life scenarios.

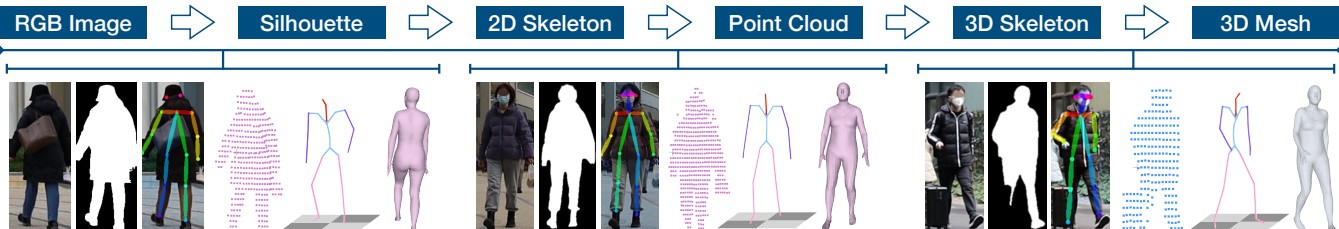

**Figure 2: Examples of diverse visual modalities and 2D/3D representations in FreeGait.**

To advance LiDAR-based gait recognition for real-world applications, we collect a novel dataset with multi-modal visual data and diverse representations in large-scale, unconstrained environments, providing a more accurate and realistic reflection of gait behaviors in real-life situations.

**LiDAR-based Applications.** LiDAR captures accurate depth information in large-scale scenarios and is unaffected by light conditions, making it popular in autonomous driving and robotics. Numerous LiDAR-based detection and segmentation methods [5, 24, 28, 41] have emerged, playing a significant role in 3D perception. Subsequently, numerous human-centric applications[7, 23, 31, 43, 48], such as pose estimation, motion capture, gait recognition, scene reconstruction, etc., have incorporated LiDAR to expand usage scenarios and improve performance of solutions by utilizing location and geometry features of LiDAR point clouds. We belive that the accurate depth-sensing capability of LiDAR can aslo denefinitly benefits in-the-wild gait recognition.

## 3 FREEGAIT DATASET

Most prevailing datasets[32, 33, 45] instruct actors to walk on predefined paths in controlled settings. While some in-the-wild gait datasets[49, 51] have been conducted in real-world contexts, they rely solely on cameras and lack 3D human dynamics. Recently, [31] claims a LiDAR-based gait dataset, but within constrained settings. In contrast, our FreeGait is a totally free-gait dataset recorded in varied real-world scenarios, from sparse to crowded pedestrian areas. It contains rich data modalities and representations, offering the potential for advanced gait recognition research in practical settings.

### 3.1 Data Acquisition and Statistics

We create a multi-modal capture device assembling a 128-beam OUSTER-1 LiDAR and a camera, calibrated and synchronized at 10fps. The LiDAR offers a 360° × 45° FOV, while the camera records at 1,920 × 1,080. Positioned at 0°, 90°, and 180° angles in scenes, three capture devices capture humans at a range of 25 meters. FreeGait includes 1,195 subjects (660 males, 535 females), with 51 recorded in low-light. The dataset is captured in real-world scenarios without any predefined paths. We select 500 subjects for training and 695 for testing, totaling 11, 921 sequences. Each subject averages 10 sequences, more than other in-the-wild datasets like Gait3D (6 sequences) and GREW (5 sequences).

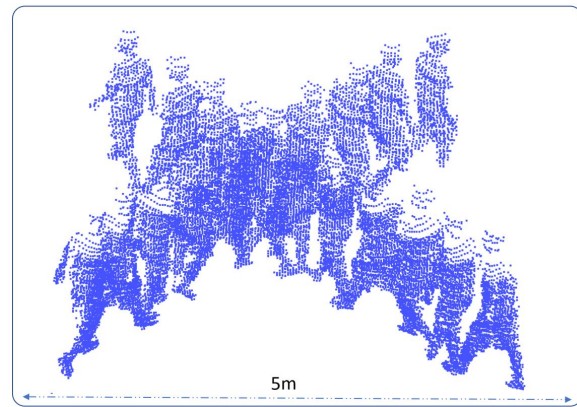

**Figure 3: Examples of predifine walking path in constained environments on SUSTech1K [31], contravenes the free gait patterns observed in real-world scenarios.**

### 3.2 Annotations

During data preprocessing, 2D silhouettes are obtained using a 2D detection method[10] and segmentation method[38], while point clouds are detected and cropped using a 3D detector[6]. In our dataset, obtaining consecutive frames of personal gait is particularly challenging due to multiple people in one scene. To achieve more accurate tracking performance, we integrate LiDAR and camera information. For point cloud-based 3D tracking, we employ the Hungarian matching algorithm, while initial ID tracking is performed using Byte-Track[47] to refine the results. Importantly, to ensure high-quality annotations, we manually correct the tracking ID for crowded scenarios with occlusions.

### 3.3 Characteristics

FreeGait offers special characteristics not available in existing LiDAR-based gait datasets. We highlight three main distinctions below. A detailed comparison with current public datasets can be found in Table 1.

*Large-scale Capture Distance.* In real-world scenarios, gait serves as a more practical biometric for long-distance person identification than face or fingerprint recognition. Existing gait datasets[31, 32] typically have a capture distance about 12 meters. With the advantages of LiDAR's long-range sensing, FreeGait is captured in diverse large-scale real-life scenarios, extending the effective human perception distance to approximately 25 meters. This large-scale

**Table 1: Comparison with public datasets for gait recognition. "Silh". and "Infr" mean silhouette and infrared, "Crowd" denotes the capture scenes involving multiple persons at the same time in uncontrolled settings, and "D&N" represents day and night during data acquisition. The dataset marked with * is no longer available.**

| Dataset | Sensor | Subject | Capture Distance(m) | Data Type | 3D Structure | Free Trajectory | Real-world | Crowd | D&N |
|---|---|---|---|---|---|---|---|---|---|
| CASIA-B [45] | Camera | 124 | N/A | RGB, Silh. | ✗ | ✗ | ✗ | ✗ | ✗ |
| CASIA-C [34] | Camera | 153 | N/A | Infrared, Silh. | ✗ | ✗ | ✗ | ✗ | ✗ |
| TUM-GAID* [11] | RGB-D & Audio | 305 | 3.6 | Audio, Video, Depth | ✗ | ✗ | ✗ | ✗ | ✗ |
| SZTAKI-LGA [2] | LiDAR | 28 | N/A | Point Cloud | ✓ | ✓ | ✗ | ✓ | ✗ |
| OU-MVLP [33] | Camera | 10,307 | 8 | Silh. | ✗ | ✗ | ✗ | ✗ | ✗ |
| GREW [51] | Camera | 26,345 | N/A | Silh., 2D/3D Pose, Flow | ✗ | ✓ | ✓ | ✓ | ✗ |
| Gait3D [49] | Camera | 4,000 | N/A | Silh., 2D/3D Pose, 3D Mesh&SMPL | ✓ | ✓ | ✓ | ✓ | ✗ |
| CASIA-E [32] | Camera | 1,014 | 8 | Silh., Infr | ✗ | ✗ | ✗ | ✗ | ✗ |
| CCPG [18] | Camera | 200 | N/A | Silh., RGB | ✗ | ✗ | ✗ | ✗ | ✗ |
| SUSTech1K [31] | LiDAR & Camera | 1,050 | 12 | Silh., RGB, Point Cloud | ✓ | ✗ | ✗ | ✗ | ✓ |
| **FreeGait** | LiDAR & Camera | 1,195 | 25 | Silh., Point Cloud, 2D/3D Pose,3D Mesh&SMPL | ✓ | ✓ | ✓ | ✓ | ✓ |

capture range enhances FreeGait's applicability in real-world gait applications.

*Real-world Scenarios.* To authentically represent human gait, FreeGait is recorded in real-world settings. Unlike prior datasets that utilize predefined trajectories in controlled environments [31, 45], FreeGait's subjects walk without any constraints, resulting in diverse and practical view variations, as well as more natural challenging factors such as various carrying and dressing, complex and dynamic background clutters, illumination, walking style and etc.

*Crowd.* Since FreeGait is captured in natural environments, frequently features multiple persons per scenario, making gait recognition more challenging than in recent SUSTech1K [31]. Notably, occlusions in crowded settings bring challenges for gait recognition due to incomplete or noisy data. While these crowded conditions pose annotation challenges, they are crucial for evaluating algorithmic robustness and advancing real-world gait-related research and applications. *More dataset statistics and examples are detailed in our supplementay.*

### 3.4 Privacy Preservation
We obey privacy-preserving guidelines. FreeGait was *constructed on campus with authorized* device placements along walkways. *Ethical discussions are detailed in the appendix.*

## 4 METHODOLOGY
### 4.1 Problem Definition
Given a LiDAR-based gait recognition dataset $\mathcal{P} = \{\mathcal{P}_i^j \mid i = 1, 2 \ldots, N; j = 1, 2, \ldots, m_i\}$ with N individuals and $m_i$ sequences for each individual $d_i$. Each sequence $\mathcal{P}_i^j$ consists of fixed $T$ frames of raw LiDAR point clouds $Y = \{y_t\}_{t=1}^T$ and projected LiDAR range views $X = \{\mathbf{x}_t\}_{t=1}^T$, where $\mathbf{y}_t \in \mathbb{R}^{N \times 3}$ is the $t_{th}$ frame of point clouds and $\mathbf{x}_t \in \mathbb{R}^{H \times W}$ is the $t_{th}$ frame of range views. For a given LiDAR point cloud of each subject, we convert each point $p_i = (x, y, z)$ to spherical coordinates and finally to 2D pixel

coordinates, using the following projection function [16]:

$$\begin{pmatrix} u_n \\ v_n \end{pmatrix} = \begin{pmatrix} \frac{1}{2}\left[1 - \arctan(y,x)\pi^{-1}\right] w \\ \left[1 - \left(\arcsin\left(zr^{-1}\right) + f_{up}\right)f^{-1}\right] h \end{pmatrix}, \quad (1)$$

where $(u_n, v_n)$ denotes range view coordinates, $(h, w)$ is the height and width of the desired range view predefined by the inherent parameters of the LiDAR. $f = f_{up}+f_{down}$ is the vertical field-of-view of the sensor, and $r = \|p_i\|_2$ is the range of each point. Then, we crop and resize the range view with each subject to a resolution of $64 \times 64$. We aim to learn a network $N_\theta(\cdot)$ that can map the inputs to feature embedding $\mathcal{F}_i^j$ to represent the corresponding individual $d_i$.

### 4.2 Overview
We propose a hierarchical multi-representation feature interaction network (HMRNet), a novel point-range gait recognition solution by taking advantage of LiDAR-projected range views and raw point clouds. Our pipeline's overview is shown in Figure 4, taking a sequence of range viewss and point clouds as input to identify individuals based on their gait. There are three main procedures in our network, including hierarchical adaptive cross-representation mapping (H-ACM), motion-aware feature embedding(MAFE), and gait-saliency feature enhancement(GSFE). We employ ResNet-like CNNs [8] and MLPs [25] to extract multi-resolution features from range views and point clouds, and fuse valuable geometric and dynamic information hierarchically. Notably, we leverage motion-aware feature embedding to explicitly model gait-related motion information from point clouds. After hierarchical multi-representation feature interaction, temporal pooling and horizontal pyramid pooling (HPP) are utilized following [8], to gather comprehensive fusion features. Before final identification, we employ gait-saliency feature enhancement module to highlight gait-informative features, benefiting from the channel-wise attention mechanism.

### 4.3 Hierarchical Adaptive Cross-representation Mapping
Raw point clouds, while rich in gait characteristics, present challenges in fine-grained feature extraction due to their sparsity and unordered nature. Conversely, range views projected from LiDAR

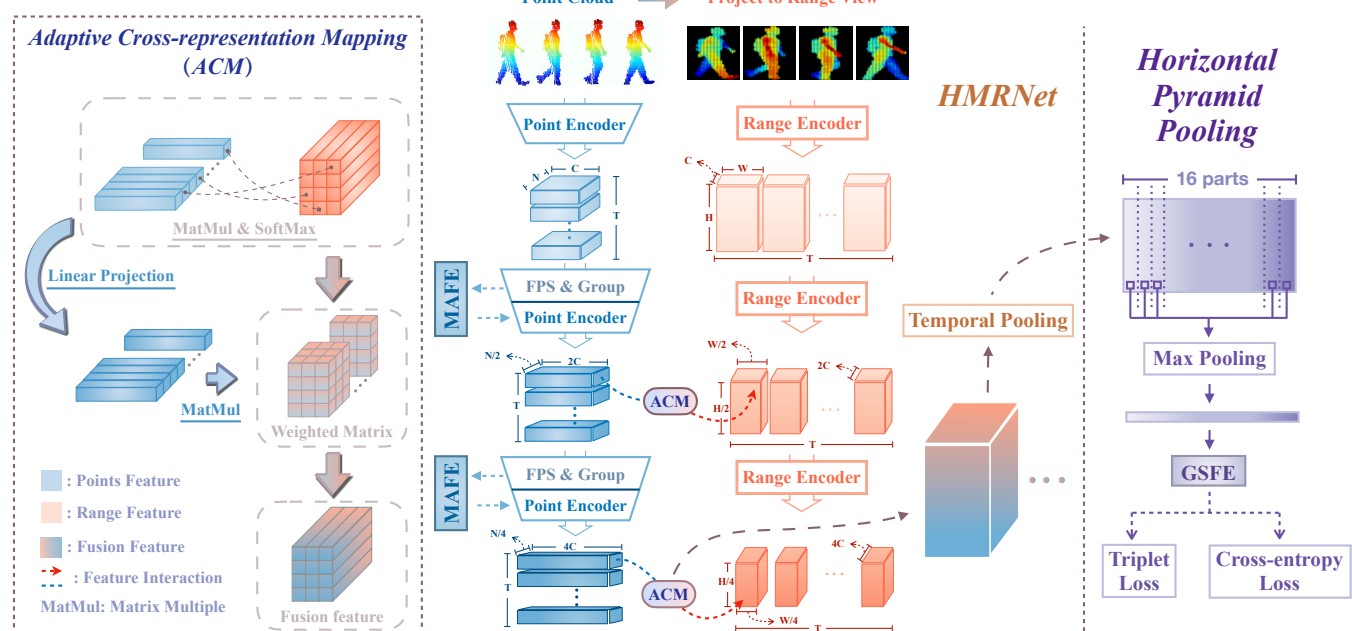

**Figure 4: The pipeline of our method. We extract dense body structure information from range views, and undistorted geometric and motion features via motion-aware feature embedding (MAFE) from point clouds. Then, adaptive cross-representation mapping module (ACM) is applied to fuse two-representation features at different levels hierarchically. Lastly, the gait-saliency feature enhancement (GSFE) module is leveraged to highlight more gait-informative features for final identification.**

point clouds offer dense and regular pixel representations, facilitating structural body feature extraction via CNNs. To combine advantages of both, we hierarchically integrate the two LiDAR representations for comprehensive features. However, fusing range view and 3D point features is non-trivial due to their inherent differences. Leveraging the transformer's capability to glean valuable global features, we utilize cross-attention to automatically search correspondences between two representation features, fuse critical gait-related information. Specifically, we structure 3D point features using range features as Query $Q$ and point features as Key $K$ and Value $V$. This dense querying from range to point yields a fusion feature rich in geometric and dynamic motion details, further benefiting distinguishing individuals. The adaptive cross-representation mapping is depicted in the left part of Figure 4. We obtain the attention map via $Q \times K$. Following softmax normalization, the attention map weights point features to amplify the range feature.

Although raw point cloud contains rich gait characteristics, their sparsity, and unordered nature brings challenges for fine-grained feature extraction. Conversely, range views projected from LiDAR point clouds offer dense and regular pixel representations, allowing for easy extraction of structural body features using CNNs. To combine advantages of both LiDAR representations, we hierarchically integrate the two representation information to obtain a comprehensive feature. However, there exists a significant gap between range view feature and 3D point feature, making it challenging to effectively fuse the two representation features. Benefiting from the transformer mechanism in acquiring valuable features in global

context, we adopt cross-attention to automatically search the correspondences between two representation features, and then fuse critical gait-related information. Especially, to organize 3D point features in a structural representation, we take range feature as Query $Q$ and point feature as Key $K$ and Value $V$ for conducting cross attention. By dense queries from range to point, we obtain a multi-representation fusion feature, consisting of rich geometric and dynamic motion information, which further benefits distinguishing individuals. Detailed operations of our adaptive cross-representation mapping process are illustrated in the left part of Figure 4. We obtain the similarity attention map by $Q \times K$. Then, the attention map is further processed by a softmax normalization and used to weight point features to enhance the range feature:

$$F_{attention} = \text{softmax}\left(\frac{QK^T}{\sqrt{d_k}}\right)V. \tag{2}$$

The final fusion feature $F_{fusion}$ is acquired through the feed-forward network (FFN) and layer normalization in transformer[37] by

$$F_{fusion} = LN(F_{attention} + FFN(F_{attention})). \tag{3}$$

Considering that different levels of features usually represent different contents, we design a hierarchical feature fusion mechanism to capture more comprehensive gait-related features. In particular, we leverage the adaptive cross-representation mapping module at two different levels to aggregate complementary low-level geometric features and high-level semantic features, as shown in the middle of Figure 4.

Table 2: Comparison with SOTA methods of gait recognition on FreeGait and SUSTech1K.

| Input | Methods | FreeGait | | | SUSTech1K | | |
|---|---|---|---|---|---|---|---|
| | | **Rank-1↑** | Rank-5↑ | mAP↑ | **Rank-1↑** | Rank-5↑ | mAP↑ |
| Silhouette | GaitSet [4] | 57.13 | 71.87 | 64.01 | 65.22 | 84.91 | 74.26 |
| | GaitPart [9] | 52.26 | 64.95 | 58.54 | 59.29 | 80.79 | 69.18 |
| | GLN [12] | 52.21 | 68.69 | 60.01 | 65.78 | 84.76 | 74.49 |
| | GaitBase [8] | 62.64 | 75.30 | 68.57 | 77.50 | 90.22 | 83.44 |
| Silhouette&SMPL | SMPLGait [49] | 53.39 | 69.25 | 60.93 | 66.34 | 85.08 | 74.95 |
| Silhouette&Key Point | SMPLGait [49] | 57.97 | 73.02 | 65.12 | 69.75 | 86.68 | 77.60 |
| 3D Key Point | GaitGraph [35] | 14.69 | 27.81 | 21.74 | 2.09 | 6.59 | 5.27 |
| Point Cloud | PointNet [30] | 42.48 | 62.48 | 52.05 | 37.31 | 65.01 | 50.11 |
| | PointMLP [25] | 57.61 | 76.68 | 66.30 | 68.86 | 90.32 | 78.55 |
| | LiDARGait [31] | 74.15 | 88.75 | 80.66 | 86.81 | 95.98 | 91.08 |
| Point Cloud | **HMRNet(ours)** | **80.76** | **93.64** | **86.53** | **90.23** | **97.54** | **93.66** |

## 4.4 Motion-aware Feature Embedding

Body part movements across successive frames are essential for effective gait recognition. However, existing LiDAR-based gait recognition methods rarely model motion information explicitly. By incorporating 3D raw point clouds that retain comprehensive pedestrian motion information, we can explicitly model gait-related dynamic information through point-wise flow. In each stage of point branch, imagine each anchor point $p_i^t$ at frame $t$ after farthest point sampling(FPS), $p_i^t$ is represented by its Euclidean coordinates $\mathbf{x}_i^t \in \mathbb{R}^3$ and a feature vector $\mathbf{f}_i^t \in \mathbb{R}^c$ from point encoder with MLPs [25]. We learn to aggregate its local geometric and dynamic motion feature by the neighboring $k$ points ($k = 16$) of $p_i^t$ in the same frame $\mathcal{N}(p_i^t)$ and nearby frame $\mathcal{N}'(p_i^{t-1})$. As shown in Figure 5, for each pair $(q_j^t, q_j^{t-1})$ in $\mathcal{N}(p_i^t)$ and $\mathcal{N}'(p_i^{t-1})$ respectively, we pass the difference of their origin Euclidean coordinates into MLPs to obtain the motion-aware feature embedding $m_j^t \in \mathbb{R}^c$ in hidden states of neighboring point $q_j^t$. Then, we get the local geometric feature with motion information for neighboring points by element-wise addition of their geometric feature $f_j^t$ and motion embedding $m_j^t$. Finally, we aggregate them by element-wise pooling and update the feature vector for every anchor point $p_i^t$. The above operations can be formulated as:

$$h(p_i^t) = \underset{q_j^t \in \mathcal{N}(p_i^t)}{MAX} \{(f_j^t + \zeta(x_j^t - x_j^{t-1})\}, \qquad (4)$$

where $\zeta$ means MLP layers, $MAX$ denotes element-wise max pooling, + is element-wise summation.

## 4.5 Gait-saliency Feature Enhancement

For the hierarchical fusion features, we use temporal pooling and Horizontal Pyramid Pooling (HPP)[8], producing a comprehensive feature vector with $p$ ($p = 16$) strips $f_s \in \mathbb{R}^c$ ($c = 512$).

Believing that different channel-wise feature maps represent various gait-related attributes with differing importance for gait recognition. To leverage this insight, we introduce a gait-saliency

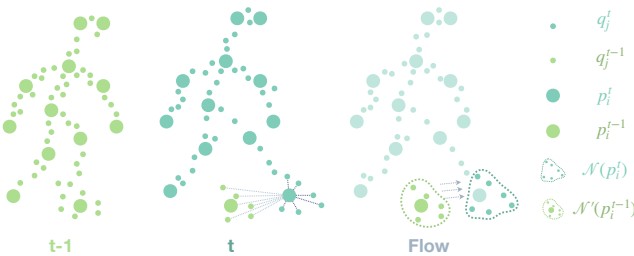

Figure 5: The procedure of point-wise flow to learn motion relation between two adjacent frames.

feature enhancement module that utilizes a channel-wise attention mechanism [13]. Our gait-saliency feature enhancement module can adaptively recalibrate channel-wise feature responses by explicitly modeling interdependencies between channels, and reweighting the input feature map to emphasize informative information. Consequently, our network efficiently highlights significant gait-related features while reducing the impact of redundant features, leading to more accurate gait recognition. Lastly, we combine gait-saliency enhancement features of various strips and obtain the final gait feature $f_\beta \in \mathbb{R}^c$ ($c = 256$) for further loss computation through a feature head with multi-FC layers.

## 5 EXPERIMENTS

### 5.1 Overview

In this section, we outline our evaluation, training, and inference details, followed by comprehensive comparison results. Ablation studies highlight our network's effectiveness, and we also analyze its robustness under cross-views, low-light conditions and various sequence length.

**Table 3: Evaluation with different attributes on SUSTech1K valid + test set. We compare our method with silhouette-based SOTA method GaitBase, LiDAR-based SOTA method PointMLP and LiDARGait.**

| Input | Methods | Probe Sequence (Rank-1 acc) | | | | | | | | Overall | |
|---|---|---|---|---|---|---|---|---|---|---|---|
| | | Normal | Bag | Clothing | Carrying | Umbrella | Uniform | Occlusion | Night | Rank1 | Rank5 |
| Silhouette | GaitBase [8] | 83.09 | 79.34 | 50.95 | 76.98 | 77.34 | 77.31 | 83.46 | 26.65 | 77.50 | 90.22 |
| Point Cloud | PointMLP [25] | 76.03 | 71.91 | 57.09 | 68.08 | 58.29 | 63.28 | 79.25 | 70.75 | 68.86 | 90.32 |
| | LiDARGait [31] | 91.91 | 88.61 | 75.27 | 88.99 | 67.55 | 81.19 | 94.73 | 90.04 | 86.81 | 95.98 |
| Point Cloud | **HMRNet(Ours)** | **92.71** | **92.34** | **79.55** | **90.27** | **83.14** | **86.19** | **95.15** | **90.35** | **90.23** | **97.54** |

## 5.2 Evaluation Protocol

For each test subject, one sequence is chosen randomly as the gallery set, with remaining sequences as the probe sets. We use Rank-k and mean Average Precision (mAP) as evaluation metrics. Rank-k measures the likelihood of finding at least one true positive in the top-k ranks, whereas mAP computes average precision across all recall levels. We present the average Rank-1, Rank-5, and mAP for the entire test set.

## 5.3 Implementation Details

For FreeGait, we resize the silhouettes to 128×88, similar to Gait3D [49]. Each LiDAR point cloud frame is resampled to $N = 256$ points via farthest point sampling(FPS), normalized by setting its center as the origin while preserving original orientations. Range views are cropped and resized to $64 \times 64$, following LiDARGait [31]. For SUSTech1K [31], we resample the point clouds to 512 points using FPS, as they have an average of 800 points compared to FreeGait's 300 points. Other input settings remain consistent with the paper. All methods, except LiDARGait and our HMRNet, are trained with Adam optimizer with a weight decay of 0.0005 and an initial learning rate of 0.001 on the two benchmarks. We employ multi-step learning rate reduction by a factor of 0.1 at the 10,000th and 30,000th iterations, with a total of 50,000 iterations. For LiDARGait and our HMRNet, we maintain the same training procedure on both benchmarks. During the training stage, each input sequence consists of 10 frames, approximating the average length of a human gait cycle. In the training stage, the loss can be formulated as follows:

$$L_{cls} = -\sum_{p=1}^{S} \sum_{c=1}^{C} gt_{s,c} \log \left( \text{softmax} \left( f_{\alpha_s} \right) \right)_c, \tag{5}$$

where $gt_{s,c}$ indicates the identity information of the $s_{th}$ strip, which equals 0 or 1. $L_{tri}$ is used to optimize the inter-class and intra-class distance, which is computed by

$$L_{tri} = \left[ D \left( f_\beta^{A_i}, f_\beta^{A_j} \right) - D \left( f_\beta^{A_i}, f_\beta^{B_k} \right) + m \right]_+, \tag{6}$$

where $A_i$ and $A_j$ are samples from the same class $A$, while $B_k$ denotes the sample from another class. $D(d_i, d_j)$ is the Euclidean distance between $d_i$ and $d_j$ and $m$ is the margin of the triplet loss. The operation $[\gamma]_+$ equals to $max(\gamma, 0)$. The overall loss function can be formulated as

$$L = \alpha L_{tri} + \beta L_{cls}. \tag{7}$$

We set the batch size to ($p = 16, k = 4$) on our FreeGait and ($p = 8, k = 8$) on SUSTech1K [31], where $p$ and $k$ denote the number of subjects and their corresponding training samples, respectively.

## 5.4 Comparison with SOTA Methods

Table 2 compares our method with state-of-the-art (SOTA) gait recognition techniques on the FreeGait and SUSTech1K datasets. Our approach outperforms exisiting sota methods LiDARGait [31], showing improvements of **6.61%** and **3.42%** in Rank-1 accuracy for FreeGait and SUSTech1K, respectively. These achievements stem from our effective combination of 3D geometry and dynamic motion gait features. Silhouette-based methods suffer from distorted body shapes in view-dependent images, impacting performance. Though *3D SMPL and KP of FreeGait and SUSTech1K* are based on the SOTA LiDAR-based mocap method [48], their accuracy is yet to be validated. We evaluated point-based gait recognition using PointNet[30] and PointMLP[25] extractors, followed by a recognition head. These point-based methods grapple with sparse representation especially in distant scenes, but PointMLP[25], leveraging local feature grouping, showcases the promise in point-based gait recognition. LiDARGait [31] mitigates point sparsity and disorder by converting them into a dense range-view representation, achieving notable results. However, this conversion can compromise the original rich 3D geometric and dynamic motion data from raw point clouds. Conversely, our HMRNet extracts dense body structures from range views and accurately captures the raw point clouds' geometry and motion. It hierarchically processes both local fine-grained and global semantic features, benefiting gait feature extraction.

We also report the results with different attributes on SUSTech1K in Table 3. Our method outperforms LiDARGait in **Bag (+3.73%)**, **Clothing (+4.28%)**, **Umbrella (+15.59%)**, **Uniform (+5.00%)** by a large margin, especially in **Umbrella (+15.59%)**. As shown in Figure 6, this is because the distorted geometric body structure from range view alone cannot bridge the gap in cases with serious appearance variance between gallery set and probe set, due to the lack of dynamic gait-related information. In contrast, our HMRNet effectively captures both dense body structures and explicitly models gait-related motion, resulting in more robust performance even in challenging scenarios.

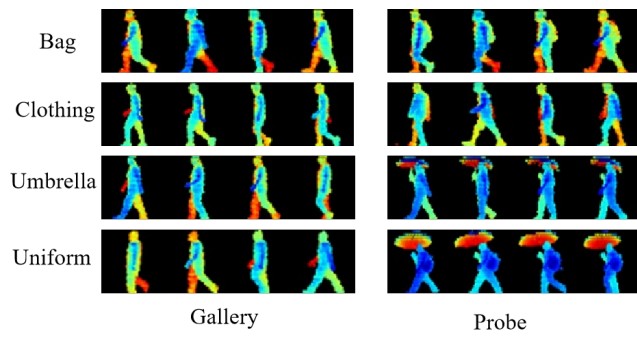

Bag

Clothing

Umbrella

Uniform

Gallery                              Probe

**Figure 6: The exemplar range views on SUSTech1K [31] in four different attributes with serious appearance variance between gallery set and probe set.**

## 5.5    Ablation for Network Design.

We verify the effectiveness of our proposed module by gradually applying the hierarchical adaptive cross-representation mapping(H-ACM) in a point-range setting, motion-aware feature embedding(MAFE), and gait-saliency feature enhancement(GSFE). The baseline is to keep only the range branch in our method, and the ablation results are shown in Table 4. **(1)** The effective integration of dense and regular body structures from range-view representation, combined with the undistorted geometric and dynamic motion information from point clouds, effectively complements one another. This results in a substantial enhancement in performance, thereby affirming the significance of our H-ACM. **(2)** MAFE explicitly models gait-related motion information through point clouds, preserving pedestrian motion details in 3D scenes and resulting in better performance. **(3)** Different features show varying importance for gait recognition across channels. Our GSFE module enhances gait-saliency features using a channel-wise attention mechanism, leading to further improvement in performance.

**Table 4: Ablation studies for network modules on FreeGait.**

| Network Module | | | | Rank-1 |
|---|---|---|---|---|
| Baseline | H-ACM | MAFE | GSFE | |
| ✔ | | | | 74.29 |
| ✔ | ✔ | | | 78.85 |
| ✔ | ✔ | ✔ | | 80.28 |
| ✔ | ✔ | ✔ | ✔ | **80.76** |

**Table 5: Rank-1 accuracy under cross views and low-light conditions on FreeGait dataset.**

| Input | Methods | Cross views | Night |
|---|---|---|---|
| Silhouette | GaitBase [8] | 30.31 | 30.83 |
| Point Cloud | LiDARGait [31] | 59.62 | 73.33 |
| Point Cloud | **HMRNet(Ours)** | **68.47** | **79.17** |

## 5.6    More Analysis of Robustness.

To assess HMRNet's robustness in real-world conditions, we test its performance under cross views and low-light scenarios. The results affirm HMRNet's robust adaptability.

*(1) Cross Views.* We select a subset from the probe set, angled approximately 90° to the gallery set for evaluation. As shown in Table 5, GaitBase's performance declines significantly in cross-views due to its reliance on appearance features, struggling with drastic view changes. LiDARGait can capture geometric body features from range-view, making it more view-tolerant. However, our method is the most robust to cross views, benefiting from the injection of dynamic gait cues from raw point clouds.

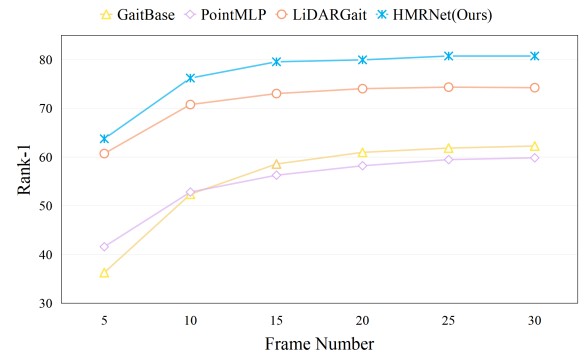

**Figure 7: Average rank-1 accuracy with increasing testing frames on FreeGait dataset.**

*(2) Low-light Conditions.* To highlight the advantage of LiDAR's insensitivity to light and the robustness of our approach, we select 27 subjects with 120 sequences collected in low-light conditions for evaluation. As shown in Table 5, the performance of 2D methods drops significantly in low-light conditions. The quality of the image can greatly impact the performance of the algorithm, resulting in poor human segmentation results. However, LiDAR-based methods, including LiDARGait and our HMRNet, can work both day and night, achieving stable performance even in the Night subset.

*(3) The Influence of Sequence Length.* We sample continuous 5-30 frames from each test set sequence for experiments and obtain the corresponding Rank-1 accuracy in Figure 7 to analyze the influence of sequence length in inference. Our method achieves remarkable results with just one gait cycle (about 10 frames) and maintains the best performance as the sequence length increases.

## 6    CONCLUSION

In this paper, we propose a new gait recognition method with an effective hierarchical multi-representation feature interaction network. We also propose a large-scale gait recognition dataset, which is collected in free environments and provides diverse data modalities and 2D/3D representations. Our method achieves state-of-the-art performance through extensive experiments. Both our novel solution and dataset can benefit further research on in-the-wild gait recognition.

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
