# OpenReview forum: "Gait Recognition in Large-scale Free Environment via Single LiDAR"
_acmmm.org/ACMMM/2024/Conference — MM2024 Oral_

### Official Review · Reviewer_4LU7 · 2024-05-11

**Rating:** 4
**Confidence:** 4

**Summary:**

The paper is focused on the task of human gait recognition using single LiDAR in real-world, unconstrained settings.
The main idea of this paper is to address the challenges in human gait recognition using multi-modal data, especially the point cloud from LiDAR.
Therefore, the main contributions include 1)  a gait recognition method that captures both dense body structure features from LiDAR range views and gait-related geometric and motion information from raw point clouds, and 2) a large-scale in-the-wild free-trajectory gait recognition dataset, i.e., FreeGait, with diverse real-world scenarios, data modalities, and data representations.
Comprehensive experiments show the effectiveness of the proposed method on both the SUSTech1K dataset and the new FreeGait dataset.

**Strengths:**

1. Overall, the writing of the paper is clear and easy to follow.
2. The proposed method is technically sound.
3. It is good to build a new dataset with unique features for the community.
4. The experimental results of the proposed method are significantly better than existing methods.

**Limitations:**

1. Figure 4 is too complex for readers. There are too many shapes with different colors, arrows with confusing directions, and groups of submodules with terms and letters. I suggest the authors carefully revise this figure. First, the overall layout and the directions of the main arrows should follow the from-top-to-left and from-left-to-right principles. Second, it should use consistent shapes for networks, features, tensors, and losses, respectively. Third, the style of lines should also be simplified. Fourth, some unimportant terms and letters like C, H, W, etc., can be removed. Last but not least, please use fewer types of colors and use pure colors instead of gradient colors.
2. Some technical details about the representations of point clouds and range views should explained. How is the range view obtained? Why is the point cloud converted to spherical coordinates? It is better to explain these with a figure.
3. In the title of this paper, the large-scale free environment is a keyword but this paper does not discuss the influence of scale/range on the performance. Because the LiDAR can capture humans at a range of 25m, I am curious what will happen if a person is captured from a close position and a far position respectively.

**Suitability:**

2

---

### Official Review · Reviewer_6gWy · 2024-05-26

**Rating:** 6
**Confidence:** 4

**Summary:**

In this paper, a gait recognition method is proposed. The authors proposed a FreeGait dataset in a large-scale free environment via a single LiDar. The proposed Gait Recognition method achieved the best reuslts in Rank-1, Rank-5, and mAP.

**Strengths:**

The authors proposed a new gait recognition dataset, capturing from the longest captured distance (25 m), and it is with the largest scale compared with the state-of-the-art datasets published in IEEE T-PAMI 2022 [32] and CVPR 2023 [31]. In addition the proposed dataset is a point cloud-based dataset. The propsod method shown in Fig. 4 is well-designed, and clearly depicted. The performance comparison shown in Table 2 achieved the best results in Rank-1, Rank-5, and mAP. The existing methods are published in CVPR 23, ICIP 21. The proposed point cloud method are compared with point cloud methods and a Silhouette-based method, and the proposed methods has the best results in all cases, as shown in Table 3 and Talbe 5.

**Limitations:**

More good and bad results from the proposed method and the state-of-the-art methods can make this paper more convincing. More discussions for the good and bad results are still missing.

**Suitability:**

3

---

### Official Review · Reviewer_2tgH · 2024-05-31

**Rating:** 4
**Confidence:** 3

**Summary:**

The article introduces "FreeGait," a new dataset and the Hierarchical Multi-representation Feature Interaction Network (HMRNet) for gait recognition using a single LiDAR sensor in large-scale, real-world environments. This technology addresses challenges like occlusions, crowd noise, and varying conditions, which are not covered by existing datasets. Captured in diverse public spaces, FreeGait includes 1,195 subjects with various demographic attributes, offering comprehensive data representations such as 2D/3D poses and 3D Mesh models. HMRNet achieves state-of-the-art performance on both FreeGait and SUSTech1K datasets, enhancing gait recognition's robustness and accuracy for applications in smart homes, healthcare, and security.

**Strengths:**

1. It has made significant progress in gait recognition technology by using a single LiDAR sensor in large-scale, real-world environments.
2. Compared to previous methods that relied on controlled environments, it shows a marked improvement in handling occlusions, crowd noise, and varying environmental conditions.
3. It helps to advance gait recognition technology further by providing a comprehensive dataset, FreeGait, which includes diverse data representations and real-world pedestrian behaviors.
4. It offers valuable insights for research in smart home automation, healthcare monitoring, and security systems, enhancing the practical applications of gait recognition.

**Limitations:**

1. The article lacks page numbers; please ensure to add them.
2. There are a few spelling mistakes in the article. In the legend of Figure 3 on page 3, "predifine" should be corrected to "predefined," and "constained" should be changed to "constrained."
3. The blank spaces below Figure 5 on page 6 and above the title of section 5.4 on page 7 are quite large, which affects the visual appearance. Please adjust the formatting accordingly.
4. Although the paper mentions the creation of the FreeGait dataset and its advantages over existing datasets, it does not provide sufficient detailed information about the annotation process or standards used to ensure data quality. In order to enhance confidence in the reliability of the dataset and encourage the research community to adopt it, the author should provide a detailed explanation of the annotation protocol, including how they address issues such as variability between and within observers and ensure consistency across the entire large-scale dataset.

**Suitability:**

2

---

### Official Review · Reviewer_ixDx · 2024-06-02

**Rating:** 5
**Confidence:** 3

**Summary:**

The paper proposed the Hierarchical Multirepresentation Feature Interaction Network (HMRNet) for robust gait recognition based on LiDAR point cloud inputs. HMRNet uses an Adaptive Cross-representation mapping (ACM) module to fuse the features of point cloud and projected range view, and extracts motion features via motion-aware feature embedding (MAFE) from point clouds. HMRNet also leverages a gait-saliency feature enhancement (GSFE) module to highlight more gait-informative features for final identification. Furthermore, the paper collected a large-scale gait dataset FreeGait in an unconstrained free walking setting. Extensive experiments on FreeGait and a previous LiDar gait dataset SUSTech1K demonstrates the effectiveness of the proposed HMRNet method.

**Strengths:**

1. The paper is in general well-written and easy to follow.
2. The paper proposed a new LiDAR-based gait dataset, which is natural, diverse, and captures the environment changes and complex crowd interactions. This dataset can be valuable for developing gait recognition algorithms in the real-world scenario.
3. The proposed HMRNet has several novel designs, including the ACM, MAFE, GSFE modules, that are shown to be helpful for improving gait recognition performance.
4. Extensive experiments demonstrate the superiority of the proposed HMRNet method compared to SOTA methods. HMRNet also shows better robustness under cross view and low-light conditions.

**Limitations:**

1. LiDAR/ point cloud-based methods might be more computationally expensive compared to silhouette-based methods. Can the authors provide some analysis on the computation cost/ speed compared to SOTA?
2. HMRNet uses both point cloud and projected range view as inputs. Have the authors tried any ablation study that uses only one input format? It will be helpful to understand the characteristics of these two input modalities.
3. Minor comments:
1) Typos: Line 259 "belive" Line 318 "contravenes" Line 442 "views".
2) In 4.1 Problem definition, N represents both the number of individuals and the number of points. Please consider using different notations.
3) There should be space between reference and text, e.g., Line 145 "methods[11]" should be "methods [11]". Similarly, there should be space between text and parentheses, e.g,. Line 446 "embedding(MAFE)" should be "embedding (MAFE)". There are many such examples throughout the paper. Please check carefully.

**Suitability:**

3

---

### Official Review · Reviewer_nHRJ · 2024-06-05

**Rating:** 4
**Confidence:** 3

**Summary:**

The paper introduces a novel method for gait recognition using a single LiDAR. The authors address the limitations of existing LiDAR-based gait datasets, which are primarily collected in controlled environments, by introducing FreeGait, a comprehensive dataset captured in large-scale, unconstrained settings.

**Strengths:**

The proposed HMRNet leverages both dense range views and raw point clouds to capture robust gait features, achieving state-of-the-art performance on both the SUSTech1K dataset and the newly introduced FreeGait dataset. The introduction of the FreeGait dataset is a significant contribution, providing a large-scale, real-world dataset that includes diverse and realistic scenarios.

**Limitations:**

1. While the paper compares its method with state-of-the-art techniques, a more detailed analysis of the computational efficiency and scalability could be beneficial.
2. The paper has some writing issues. For instance, the first two paragraphs on page five contain a lot of repetitive information.
3. In Figure 2, several images are actually identical and do not effectively demonstrate what each subheading represents.

**Suitability:**

3

---

### Official Review · Reviewer_oZce · 2024-06-10

**Rating:** 4
**Confidence:** 3

**Summary:**

The paper presents an approach to gait recognition in free environments using a single LiDAR sensor. The authors introduce the Hierarchical Multirepresentation Feature Interaction Network (HMRNet), which is designed to robustly recognize gait by leveraging both the dense range views and the raw point clouds from LiDAR data. They also introduce a new dataset, FreeGait, which contains a diverse set of real-life scenarios and is aimed at facilitating research in LiDAR-based gait recognition. The paper demonstrates the good performance on the SUSTech1K dataset and the newly introduced FreeGait dataset.

**Strengths:**

1. The introduction of the FreeGait dataset is a good contribution, providing a large-scale, diverse, and realistic set of gait data that can drive further research in the field.
2. The HMRNet architecture effectively combines the strengths of dense range views and raw point clouds, potentially offering a more comprehensive feature set for gait recognition compared to methods relying on a single data representation.
3. The paper includes a thorough experimental evaluation, showcasing the method's performance across various conditions and demonstrating its robustness.

**Limitations:**

1. Detail. Line 433, viewss changed to views.
2. Figure 4 looks a little confused and unclear.
3. Line 833 is out of the margin.
4. The built dataset contains annotated information for different views and lights, i.e., the relevant statistics should be given.

**Suitability:**

2

---

### Meta-Review · Area_Chair_n8a7 · 2024-07-01

**Recommendation:** Accept (Oral)
**Confidence:** 3

**Metareview:**

This paper presents a gait recognition method utilizing an effective hierarchical multi-representation feature interaction network, accompanied by a dataset. While some details still need refinement, the overall quality of the writing is commendable.